# A Practical Multidisciplinary Approach to Identifying Interstitial Lung Disease in Systemic Autoimmune Rheumatic Diseases: A Clinician’s Narrative Review

**DOI:** 10.3390/diagnostics14232674

**Published:** 2024-11-27

**Authors:** Viorel Biciusca, Anca Rosu, Sorina Ionelia Stan, Ramona Cioboata, Teodora Biciusca, Mara Amalia Balteanu, Cristina Florescu, Georgiana Cristiana Camen, Ovidiu Cimpeanu, Ana Maria Bumbea, Mihail Virgil Boldeanu, Simona Banicioiu-Covei

**Affiliations:** 1Department of Internal Medicine-Pneumology, Faculty of Medicine, University of Medicine and Pharmacy of Craiova, 200349 Craiova, Romania; biciuscaviorel@gmail.com (V.B.); sorina_stan@cloud.com (S.I.S.); 2Department of Internal Medicine-Rheumatology, Faculty of Medicine, University of Medicine and Pharmacy of Craiova, 200349 Craiova, Romania; anca.rosu@umfcv.ro (A.R.); simona219@yahoo.com (S.B.-C.); 3Institute of Diagnostic and Interventional Radiology, Goethe University Hospital Frankfurt, Theodor-Stern-Kai 7, 60596 Frankfurt am Main, Germany; teodorabiciusca96@gmail.com; 4Department of Pneumology, Faculty of Medicine, Titu Maiorescu University, 031593 Bucharest, Romania; 5Department of Internal Medicine-Cardiology, Faculty of Medicine, University of Medicine and Pharmacy of Craiova, 200349 Craiova, Romania; tohaneanu67@yahoo.com; 6Department of Radiology and Medicine Imaging, Faculty of Medicine, University of Medicine and Pharmacy of Craiova, 200349 Craiova, Romania; gia_toma@yahoo.com; 7Clinical Hospital of Infections Diseases “Victor Babes”, 200515 Craiova, Romania; cimpeanuovidiu94@yahoo.com; 8Department of Medical Rehabilitation, Faculty of Medical Assistance, University of Medicine and Pharmacy of Craiova, 200349 Craiova, Romania; anamariabumbea@yahoo.com; 9Department of Immunology, Faculty of Medicine, University of Medicine and Pharmacy of Craiova, 200349 Craiova, Romania; mihail.boldeanu@umfcv.ro

**Keywords:** multidisciplinary approach, systemic autoimmune rheumatic diseases, interstitial lung disease, early diagnosis

## Abstract

Interstitial lung disease (ILD) is one of the common and potentially lethal manifestations of systemic autoimmune rheumatic diseases (SARDs). ILD’s prevalence, clinical patterns, imaging, and natural history are variable. Each of the representative diseases—systemic sclerosis (SSc), idiopathic inflammatory myopathies (IIMs), rheumatoid arthritis (RA), Sjӧgren’s syndrome (SjS), mixed connective tissue disease (MCTD), systemic lupus erythematosus (SLE)—have distinct clinical, paraclinical and evolutionary features. Risk factors with predictive value for ILD have been identified. This review summarizes, from the clinician’s perspective, recent data from the literature regarding the specificity of ILD for each of the autoimmune rheumatic diseases, with an emphasis on the role of the multidisciplinary team in early diagnosis, case management, as well as the particularities of the clinical approach to the progressive phenotype of ILD in SARDs.

## 1. Introduction

Interstitial lung disease (ILD) is a common manifestation of systemic autoimmune rheumatic diseases (SARDs), characterized by pulmonary interstitial damage demonstrated by high-resolution computerized tomography (HRCT) and established after excluding other causes of pulmonary damage (such as infections, drugs, toxins, malignant diseases) [1]. The term previously used to describe this condition, often found in the literature, is ILD associated with connective tissue disease (CTD-ILD). SARDs that are associated with ILD include systemic sclerosis (SSc), rheumatoid arthritis (RA), idiopathic inflammatory myopathies (IMMs, including polymyositis, dermatomyositis, antisynthetase syndrome, immune-mediated necrotizing myopathy), Sjӧgren’s syndrome (SjS), mixed connective tissue disease (MCTD), and lupus systemic erythematosus (SLE). Although the incidence rate of SARD-ILD is currently estimated at 15–20%, the condition is underdiagnosed [2]. A multidisciplinary approach to SARD-ILD is essential for accurate diagnosis and appropriate management of these complex conditions [3]. The steps in achieving this goal must take into consideration several features such as the complexity of clinical and paraclinical evaluation in SARDs, the development of personalized management and long-term monitoring following interdisciplinary team meetings [4]. Early diagnosis of SARD-ILD is critical for timely treatment [5]. Initial evaluation requires a comprehensive clinical exam, focusing on detecting ILD risk. Bilateral basal crackles suggest possible ILD, warranting pulmonary function tests (spirometry, lung volumes, diffusion capacity) and HRCT for confirmation. HRCT is the current reference test for the diagnosis of SARD-ILD [6]. Pulmonary function tests (PFTs) help assess how SARD-ILD affects lung function, with spirometry, lung volumes, and diffusing capacity of the lungs for carbon monoxide (DLco) being key diagnostic tools [7]. Pathological diagnosis is reserved for cases with non-diagnostic imaging, suspected infections, or drug toxicity. Confirming SARD-ILD diagnosis and assessing lung damage require a broad range of tests. Serological, imaging, functional, pathological, and laboratory results must be interpreted jointly, correlating pulmonary imaging findings with patient symptoms and functional test outcomes [8]. Multidisciplinary meetings are essential for creating personalized treatment plans, combining each team member’s expertise [9]. Assessment by a multidisciplinary team (MDT) has been proposed as the standard of excellence in the management of ILD [10]. Managing SARD-ILD requires long-term monitoring to detect complications and adjust treatment based on disease progression. Despite antifibrotic therapy, the unpredictable course of SARD-ILD imposes regular evaluation of disease and treatment response [11]. This approach ensures ongoing care tailored to the patient’s needs. Additionally, educating patients and families about the disease and treatment options improves adherence and quality of life, ultimately enhancing disease management and prognosis [12]. Achieving these objectives through a multidisciplinary approach in early SARD-ILD diagnosis can improve disease management and prognosis, providing patients with the support and care they need [13]. The frequency of SARD-ILD is increasing due to improved diagnostics like HRCT, but the true incidence remains unclear, with global variation in reports. Many uncertainties persist (exposure to high and repeated doses of radiation is of concern, and the utility of serum biomarkers remains commercially unfeasible), requiring significant expertise and close collaboration within the MDT [14]. Given the disease’s unpredictable progression and severity, developing a diagnostic and monitoring protocol with clear team roles is essential for rapid diagnosis [15]. This article aims to update recent research and provide specialists, including primary care physicians, with valuable information on a relevant but under-discussed topic. Studies emphasize the importance of managing SARD-ILD to respond to unmet needs for improving screening and monitoring [16].

## 2. Diagnostic Tools

Pulmonary involvements are severe complications in patients with SARDs, linked to higher mortality and morbidity rates. Among SARD-related respiratory complications, ILD poses the most significant implications, often leading to respiratory failure and premature death [17]. Management of the disease requires early diagnosis and assessment of severity and progression risk. Clinical guidelines for SARD-ILD from the US [18] and Europe [19] outline the need for disease identification and its proper management. Studies have developed algorithms [20] that consider specific risk factors, the utility of HRCT, and the importance of an MDT. This collaboration establishes that combining clinical exams, HRCT, PFTs, and specific ILD biomarkers effectively identifies pulmonary interstitial involvement in patients [21]. SARD-ILD diagnosis requires a complete and correct clinical examination, which includes general symptoms and non-specific respiratory symptoms. The clinical examination is complemented by serological tests, which help establish the diagnosis of SARD-ILD and imaging studies that are essential for diagnosing, identifying the disease and assessing improvement. PFTs aim to identify the type of ventilatory dysfunction and the severity of respiratory failure. Histological or cytological examination of the bronchoalveolar lavage fluid is performed when imaging and functional investigations are inconclusive [15].

### 2.1. Clinical Examination

The complete clinical examination will highlight the constitutional, cutaneous, osteoarticular, respiratory, cardiac, gastroenterological, hematological, and neurological manifestations that patients may present from the onset of SARDs. In addition to constitutional symptoms (fever, physical asthenia, loss of appetite, weight loss), the clinician will encounter respiratory signs (unexplained breathlessness, persistent dry cough, exertional dyspnea, and exercise intolerance). Other possible symptoms are chest pain, palpitation, tachypnea, and hemoptysis. The patient’s history will also reveal some of the risk factors for the occurrence of SARD-ILD (Table 1).

The objective pulmonary exam will reveal changes in lung resonance and the presence of adventitious sounds, such as bilateral crackles at the base of both lungs, detected during slow and deep breaths, commonly to known as “velcro crackles”.

### 2.2. Serological Testing

Serological testing is also part of the rheumatologist’s diagnostic work, as there are serological markers or autoantibody profiles that strongly suggest a specific SARD. Serological profiles help clinicians focus on specific organs during assessments. Expanded autoantibody panels, like the myositis-specific autoantibody (MSA) panel, improve the identification of SARDs [32]. Serological testing may follow pneumatological evaluation and HRCT findings, such as a non-specific interstitial pneumonia-organizing pneumonia (NSIP-OP) overlap pattern seen in IIM, and anti-synthetase syndrome (ASS) [57]. Importantly, serology results should always be interpreted within the clinical context, often requiring input from rheumatologists or immunologists.

### 2.3. Imaging Examination

The standard chest radiograph has little diagnostic value for ILD. The 2023 ACR Guidelines for the Screening and Monitoring of Interstitial Lung Disease in People with Systemic Autoimmune Rheumatic Diseases do not recommend standard chest radiographs for ILD diagnosis in SARD patients [2]. Radiologically, interstitial syndrome shows diffuse bilateral opacities, including fine reticular, reticulo-nodular, micronodular, alveolar exudative, or retractile opacities.

HRCT is essential for ILD diagnosis, playing an important role in the differentiation and etiological diagnosis of interstitial syndromes. The clinical form of ILD can be suggested based on age and lesion patterns. HRCT reveals combinations of four lesion patterns (linear, nodular, alveolar/consolidation, cystic), with lesion type and location being very important for diagnosis. HRCT plays a key role in suspected SARD-ILD, classifying disease patterns [58], identifying characteristic features, rapidly assessing the extent of ILD, and detecting complications such as infections, drug-induced pneumonitis, exacerbations, or malignancy [59,60]. The main HRCT imaging and pathological findings in SARD-ILD are summarized in Table 2.

In patients with SARDs we encounter several forms of ILD. NSIP has a more homogeneous appearance without honeycombing, with symmetrical areas of ground glass opacities and fibrosis, which is distributed diffusely or subpleurally. The exception is RA in which the usual interstitial pneumonia (UIP) predominates and predicts poorer outcome. UIP presents a heterogeneous distribution, with honeycombing (subpleural cysts) and predominant fibrosis at the bases of the lungs, which are associated with inflammatory and fibrotic lesions of different ages [78]. The HRCT features and axial images described in patients with SARD-ILD are presented in Table 3.

### 2.4. Pulmonary Function Tests

PFTs evaluate lung function, identifying respiratory, neuromuscular, or cardiovascular disorders. PFT components include spirometry, lung volume measurement, DLco, arterial blood gas analysis (ABGA), and exercise testing [79].

In SARD-ILD, spirometry reveals restrictive ventilatory dysfunction with reduced vital (VC) and forced vital capacities (FVC) but a normal or increased forced expiratory volume in the first second (FEV1) to FVC ratio [80]. Obstructive dysfunction, which occurs less frequently in ILD, may indicate the presence of inflammation, as in chronic obstructive pulmonary disease (COPD) or bronchiectasis [73]. PFTs help guide treatment, monitor disease progression, and may indicate the need for further investigations or adjustments in therapy [73,81,82]. Ideally, we would be able to identify from spirometry patients who need additional pulmonary function tests, including lung volume and Dlco [83,84]. Plethysmography confirms lung volume restrictions, with total lung capacity (TLC) below the lower limit of normal (LLN) indicating a true restrictive pattern. Severity is graded based on TLC and FVC, ranging from mild to very severe [85]. A “non-specific restrictive pattern” is seen in 10% of patients with abnormal spirometry and normal TLC [86]. Approximately 50–60% of patients have bronchial obstruction. Measuring lung volumes and chest imaging helps identify pulmonary dysfunction causes in SARD-ILD patients [86]. ILD patients often have mixed ventilatory disorders with both restrictive and obstructive patterns, including COPD [87]. DLco, a key component of PFTs, assesses gas exchange in SARD-ILD patients. Low DLco values and DLco/CVF ratio suggest reduced lung volumes such as in ILD, emphysema, and pulmonary vascular diseases. DLco severity classification includes normal Dlco (>75% of the predicted value), mildly reduced DLco (60% to 75% of the predicted value), moderate (40%–60% of the predicted value), and severe (<40% of the predicted value) [88]. ABGA helps detect hypoxemia, hypercapnia, and respiratory failure [89]. Regular assessments are essential for tracking complications like chronic respiratory failure (CRF), and pulmonary hypertension [80,90]. The 6 min walk test (6 MWT) aids to asseess prognosis and lung transplant eligibility [91]. Desaturation in the exercise test is a strong mortality predictor [64]. Performing the 6 MWT every 3–4 months is recommended for SARD-ILD patients [92].

### 2.5. Pathological Exam

In patients with SARD-ILD, lung biopsies show fibro-inflammatory changes in various lung compartments, including airways, alveolar septa, pleura, and vasculature [93]. Surgical lung biopsy, including video-assisted thoracoscopic biopsy, aids diagnosis and shows specific ILD histological patterns (Table 2) [94].

### 2.6. Bronchoalveolar Lavage

Cytological analysis of bronchoalveolar lavage (BAL) fluid is useful in managing various lung diseases, particularly ILD [95]. In healthy, non-smoking adults, BAL fluid typically contains fewer than 13 million cells, primarily alveolar macrophages (over 80%), with smaller percentages of lymphocytes, neutrophils, eosinophils, and mast cells [96]. BAL analysis assesses lung inflammation, detects SARD-specific cells like activated T lymphocytes [97], identifies infections [98], and monitors treatment response [99]. Regular evaluations help adjust therapy by monitoring changes in cellular composition or biochemical markers.

### 2.7. Serum Markers

Serum markers used in diagnosing and evaluating the prognosis of SARD-ILD are classified into four categories. Key markers for alveolar dysfunction include Krebs von den Lungen-6 (KL-6), mucins (MUC-1, MUC-4, MUC-5, MUC-16), tumor markers (CA 15-3, CA 125, CA 19-9), surfactant proteins (Sp-A, Sp-D), and club cell secretory protein (CC16). KL-6 is a valuable biomarker for the early diagnosis and severity assessment of ILD, especially in SSc-ILD [100]. Research shows an increase in KL-6, along with the tumor markers CEA, CA153, and CA125, in patients with ILD, indicating its non-invasive diagnostic potential [101]. Elevated CA153 and CYFRA21-1 levels signal increased ILD risk [102], while elevated CA125 can help assess ILD risk in RA patients [103]. Increased levels of SP-A and SP-D also support early detection of SSc-ILD [104], and combining SP-D with imaging improves diagnosis [105]. Additional biomarkers such as chemokine CCL18, protein YKL-40, matrix metalloproteinase (MMP-7) and CC16 further improve risk assessment for ILD progression [106,107,108]. Biomarkers of immune dysfunction in ILDs include cytokines or chemokines like IL-8, C-C motif ligand-2 (CCL2), CC chemokine ligand 18 (CCL18), CXC motif chemokine ligand 1 (CXCL1), C-X-C motif chemokine ligand 10 (CXCL10), C-X-C motif chemokine ligand 12 (CXCL12), C-X-C motif chemokine 13 (CXCL13), TNF-α, and others involved in innate and acquired immunity. Elevated IL-8 is observed in SSc-ILD, RA-ILD, and DM/PM-ILD [109,110], while CCL2 predicts mortality in SSc-ILD and differentiates ILD from infections [111,112]. TNF-α is elevated in juvenile dermatomyositis, and IL-6 indicates early progression in DM-ILD [113]. CCL18 predicts progressive fibrosing in IIM-ILD [114], and YKL-40 aids in diagnosing SARD-ILD [115]. CXCL1 is linked to SSc-ILD activity [116] and potential improvement with rituximab [117], while CXCL10 levels are elevated in SSc-ILD [118] and RA-ILD [119]. CXCL13 is important for monitoring SjS [120] and plays a role in lung cancer diagnosis [121], with studies showing upregulated CXCL13 in 90% of non-small-cell lung cancer cases [122], linked to neovascularization and tumorigenesis [123]. MMP-3, MMP-7, and MMP-9 are biomarkers for extracellular matrix remodeling and fibroblast proliferation in SARDs, including SSc-ILD [124]. Elevated serum levels indicate lung damage, with MMP-7 aiding in assessment of disease progression, while MMP-9, SP-D, and VEGF assist in assessment of severity [125]. Genetic biomarkers for early SARD diagnosis include MUC5B promoter polymorphism, Toll-interacting protein (TOLLIP) and Toll-like receptor 3 (TLR3 receptor) polymorphisms, and telomere-related gene (TRG) mutations. MUC5B polymorphism is a key marker for early RA-ILD detection, particularly in UIP-pattern patients [55,126], while PRR polymorphisms indicate infection susceptibility. Further research is needed to validate these genetic markers [127].

### 2.8. Multidisciplinary Team

An MDT plays an essential role in early diagnosis and management of SARD-ILD [128], involving pulmonologists, rheumatologists, radiologists, cardiologists, and pathologists [129]. Accurate diagnosis requires correlation between clinical, radiological, serological, and functional data [15]. Clinicians lead initial assessments through exams, medical history, and serological tests [130], while chest radiologists use HRCT for detecting ILD [131]. Collaboration in MDT meetings enhances case debate, allowing for identification of early disease and complications, with cardiologists playing a key role in detecting pulmonary hypertension (PH) [132]. Echocardiogram, cardiopulmonary stress test, and atrial natriuretic peptide/brain natriuretic peptide (BNP/NT-proBNP) have been identified as screening tools for PH in SARD-ILD. Right heart catheterization was considered essential to confirm PH [133]. Cases with pathological evidence should be reviewed in MDT discussions, and a working diagnosis is formulated [134]. This collaborative approach optimizes the treatment and patient outcomes [19].

## 3. Early Diagnosis SARD-ILD

The early diagnostic approach to SARD-ILD includes regularly screening high-risk patients for ILD [1]. The American College of Rheumatology (ACR)/American College of Chest Physicians (CHEST) Guidelines for Screening and Monitoring of Interstitial Lung Disease in Individuals with SARDs conditionally recommend screening with PFTs and chest HRCT, while avoiding tests such as bronchoscopy, chest radiography, and surgical lung biopsies [18]. Screening should be tailored to individual risk factors for ILD, such as high titers of anti-CCP or RF positivity in RA-ILD, early diffuse skin disease in SSc-ILD patients, and specific antibodies in IIM-ILD [135]. Physical examination, PFTs, and HRCT are recommended for high-risk patients. Combining PFTs and HRCT scans provides complementary insights, making it preferable. Screening frequency should be determined based on ILD risk, with annual PFTs for patients with SARDs and more frequent assessments (within the first 4–5 years) for those with symptoms or additional risk factors [136].

### 3.1. Early Diagnosis of SSc-ILD

Early diagnosis of SSc-ILD involves comprehensive screening of newly diagnosed SSc patients [136]. Initial assessment includes a detailed clinical examination by a rheumatologist, HRCT, and PFTs [1,134,137,138,139]. Clinical examination may reveal respiratory symptoms and signs of SSc-ILD. HRCT is preferred for its sensitivity in detecting interstitial changes specific to ILD, while PFTs establish baseline respiratory function values. However, in many cases, although HRCT indicates ILD, FVC and DLco remain normal, limiting these parameters’ effectiveness as screening tools [1]. Patients without baseline HRCT evidence of SSc-ILD are stratified by ILD risk and monitored using a specific algorithm that includes regular PFTs, HRCT, and occasionally the 6 MWT to assess lung function (Figure 1). Those at high risk for ILD undergo PFTs every 6–12 months and HRCT annually during the first few years or as respiratory changes occur. In case of low risk, PFTs are performed annually, and HRCT only in the presence of new symptoms. Patients diagnosed with SSc-ILD are monitored for pulmonary hypertension (PH), especially if they present with unexplained dyspnea [134]. SSc-PH screening includes transthoracic echocardiography, DLco and biomarkers (BNP/NT-proBNP), as early identification of SSc-PH is essential given the availability of new and effective drugs [140,141]. Periodic evaluation monitors the activity and severity of SSc-ILD to promptly detect forms of PPF-SSc.

### 3.2. Early Detection of MCTD-ILD

Screening for MCTD-ILD is very important due to the increased risk of pulmonary involvement, which can lead to a severe prognosis [43,142,143]. Initial evaluation consists of periodic clinical examinations, HRCT, and PFTs. HRCT can detect lung lesions corresponding to NSIP, UIP, and OP patterns, even in the absence of symptoms [144,145]. PFTs, including FVC and DLco, are essential for the assessment of respiratory function [10]. Some studies have shown that, over ten years, FVC and TLC values remained stable in MCTD, while DLco decreased, correlating with active ILD, demonstrating its superior sensitivity in detecting the disease [146,147]. After initial assessment, patients are stratified according to risk factors for MCTD-ILD. The frequency of screening depends on the severity of the disease and symptoms. Given that the SSc-like phenotype of MCTD is frequently associated with ILD, patient monitoring should follow an algorithm similar to that for SSc patients. Patients at high risk will be evaluated by PFTs every 6–12 months and exercise desaturation test will be performed every 3–6 months. HRCT reassessment is required annually for the first 3–4 years or whenever respiratory changes are observed. Low-risk patients will have annual clinical examinations and PFTs, with HRCT performed only if significant abnormalities occur.

### 3.3. Early Detection of IIM-ILD

Screening for IIM-ILD is essential because ILD can be a serious and frequent complication. ILD is common in the ASS of IIM, varying between chronic stationary forms and rapidly progressive phenotypes of ILD. Anti-MSA and anti-MAA antibodies correlate with different clinical phenotypes [148]. Screening consists of clinical and serological evaluation by a rheumatologist, including analysis of the autoantibody profile (anti-MDA5 and anti-synthetase) to identify symptomatic or high-risk patients. Myositis antibody panels that test for many autoantibodies are useful to stratify suspected IIM patients for ILD and also for cancer risk. For patients with specific antibodies (anti-MDA5 and anti-synthetase), HRCT is the gold standard for the detection of fibrotic lesions [149,150]. PFTs monitor respiratory capacity, and the exercise desaturation test assesses lung capacity under exercise conditions. The frequency of reassessments varies according to the individual risk. In patients with risk factors for ILD, if the initial HRCT is normal, it will be repeated every 1–2 years. If respiratory symptoms occur or there is deterioration of lung function, HRCT may be repeated more frequently at 6–12-month intervals. If changes suggestive of ILD occur on HRCT, then the patient will be monitored clinically, functionally, and by imaging with higher frequency. PFTs should be performed every 3–6 months to monitor disease progression. These tests will be repeated every 3–6 months in patients at high risk of IIM-ILD or those who are symptomatic. In patients without risk factors, PFTs can be performed at 6–12-month intervals. The exercise desaturation test will be performed every 6–12 months to assess lung capacity under exercise conditions. Significant desaturation with exercise will require repeat PFTs and HRCT to reassess lung involment.

### 3.4. Screening for ILD in Patients with SjS

Screening for SjS-ILD is essential because the real prevalence of this condition is often underestimated [151]. Clinical assessment performed by the rheumatologist helps identify pulmonary symptoms such as exertional dyspnea and cough. In asymptomatic patients, the 2021 Consensus Guidelines for the evaluation and management of Sjögren’s lung disease recommend screening all patients with Sjögren’s at baseline with chest radiography and recommend consideration of PFTs [152,153]. If there is clinical suspicion of pulmonary involvement, HRCT can confirm or exclude presence of ILD and detect possible lymphoproliferative complications, such as amyloidosis or bronchial-associated lymphoid tissue (BALT) lymphoma. HRCT is used to define the pattern of SjS-ILD, where NSIP is the most common pattern observed in SjS-ILD, followed by UIP, OP, and LIP [154]. In patients with risk factors for ILD, PFTs reveal restrictive ventilatory dysfunction, with a reduction of TLC and FVC, with a normal FEV1/FVC ratio [155,156]. The frequency of screening depends on each patient’s risk. In those with risk factors, HRCT is recommended initially and every 2–3 years thereafter [6]. Following the Consensus Guidelines for the Evaluation and Management of Pulmonary Disease in Sjögren’s Disease, PFTs are repeated every 3–6 months, especially in the first 1–2 years [152]. Patients without risk factors require less frequent reevaluations. This regular screening allows early detection of SjS-ILD, facilitating rapid therapeutic intervention and improving the prognosis for SjS patients.

### 3.5. Early Detection of RA-ILD

Early detection and monitoring of patients with RA-ILD are essential for prompt treatment decision, given that lung function impairment may be significant at the time of diagnosis [157]. Initial assessment of pulmonary involvement should include clinical examination, PFTs, and HRCT [18,158,159]. The clinical examination, performed by the rheumatologist, can identify symptoms of ILD (Figure 2). Because chest radiography has low sensitivity and specificity for RA-ILD, it will not be performed routinely [160]. HRCT and PFTs are performed initially in patients with respiratory symptoms, and in some cases, these tests can help identify pulmonary involvement [161]. After initial assessment, patients are screened for risk of developing RA-ILD. Those with risk factors will be monitored annually through PFTs. If the initial screening is negative, PFTs will be repeated annually. If ILD is confirmed, PFTs will be repeated every 3–12 months for the first year, with the frequency adjusted thereafter according to disease progression. HRCT reassessment should be performed every 2 years or whenever respiratory changes occur. Low-risk patients will have annual evaluations by PFTs and HRCT if necessary.

### 3.6. Early Detection of SLE-ILD

Early detection of SLE-ILD is important to prevent severe complications and improve prognosis [162]. Screening includes a careful clinical assessment focused on the respiratory system, considering symptoms such as chest pain, dyspnea, cough, hemoptysis, and reduced exercise tolerance. These symptoms may indicate the need to investigate the underlying lung disease by HRCT and PFTs. HRCT is considered the gold standard for the early diagnosis of SLE-ILD, as it confirms the presence of the disease and determines its extent. In high-risk patients, HRCT, FVC, and DLco tests are performed to assess the extent of lung lesions and progression of ILD. Antibodies such as anti-Scl-70, anti-La/SSB, and anti-U1 RNP are also important because they are associated with an increased risk of SLE-ILD [153,163]. The frequency of screening varies according to the severity of the disease; a general clinical examination should be performed every 6–12 months. The initial HRCT is recommended at diagnosis, and in case of a normal result, it is repeated every 1–2 years. PFTs are performed every 6–12 months, increasing the frequency in the presence of symptoms [164]. In patients without risk factors, assessment may be less frequent.

## 4. Monitoring SARD-ILD Patients

The monitoring of SARD-ILD patients aims to assess disease activity and severity. Over time, careful monitoring helps identify the phenotypic and genotypic characteristics that define progressive pulmonary fibrosis (PPF) within each SARD-ILD. PPF in ILDs is responsible for irreversible lung damage, increased morbidity, and mortality [165]. Originally, Cottin proposed the following diagnostic criteria for PPFin ILDs: a relative decrease in FVC ≥ 10%, a relative decrease in the DLco ≥ 15%, or a relative decrease in FVC ≥ 5% but <10%, in combination with worsening symptoms or radiographic findings in the previous 24 months [166]. More recently, the INBUILD trial defined the PPF in ILDs based on the fulfillment of at least one of the following criteria, over 24 months, despite standard treatments (except nintedanib or pirfenidone): an estimated relative decrease in FVC ≥ 10%; a predicted relative decrease in FVC of ≥5% to <10%, associated with either worsening respiratory symptoms or extension of fibrosis seen on chest HRCT; or a combination of worsening respiratory symptoms and extension of fibrosis on HRCT [167]. Currently, the hallmarks of PPF in ILDs, and thus of SARD-ILD, are progression of pulmonary fibrosis, worsening of symptoms, decline in lung function, and deterioration of the quality of life [168]. In medical practice, it is ideal to detect PPF in SARD-ILD as soon as possible.

### 4.1. Predictive Factors for SARD-ILD Progressive Fibrosing Phenotype

Monitoring will primarily target patients identified as having predictive risk factors for PPF (Table 4).

The ACR/CHEST Guidelines for the screening and monitoring of ILD in individuals with SARD conditionally recommended monitoring for ILD progression with PFTs and HRCT over PFTs alone. An additional conditional recommendation is for added monitoring with ambulatory desaturation testing. Also, the ACR/CHEST Guidelines conditionally recommend against monitoring with bronchoscopy, chest radiography, and 6 MWT [18]. The progression of SARD-ILD may vary depending on the condition and is influenced by the presence of factors predictive of a progressive fibrosing phenotype. Monitoring ILD progression in symptomatic patients with SARD-ILD should be a joint effort between rheumatology, pulmonology, and imaging specialists. Periodic monitoring with clinical assessment and PFTs is essential in providing useful information over time, especially for patients with respiratory symptoms such as cough or dyspnea. HRCT scans are also recommended to assess the extent and pattern of pulmonary involvement, with their frequency determined by symptoms, functional changes, and radiation exposure risk.

A combination of PFTs and HRCT is more suitable for optimal monitoring of ILD progression. Ambulatory exercise desaturation testing is recommended based on symptoms to assess the need for oxygen therapy. Chest radiography and the 6 MWT are not indicated due to low sensitivity and the influence of other health conditions. Bronchoscopy is not recommended for routine monitoring, due to the risks to which patients are exposed, but may be useful in specific cases, such as the evaluation of pulmonary infections in patients with atypical imaging and symptoms [18]. The frequency of monitoring should be individualized, with PFTs suggested quarterly or every 6 months in the first year among patients with IIM-ILD and SSc-ILD and twice a year or annually once SARD-ILD is considered stable. For patients with SjS-ILD, MCTD-ILD, and RA-ILD, the use of PFTs is recommended initially quarterly and then annually or even more frequently if the ILD is considered unstable. HRCT scans should be performed when clinically or functionally indicated. Exercise desaturation testing frequency varies from quarterly to annual intervals based on respiratory symptom severity.

### 4.2. Monitoring SSc-ILD Patients for PPF Diagnosis

Patients with SSc-ILD will be monitored to assess the activity and severity of interstitial lung disease. Additionally, monitoring of these patients aims to detect PPF phenotype as early as possible.

Risk assessment for PPF in SSc-ILD will be based on the presence or absence of specific predictors for this severe form of the disease [196], which is essential for identifying patients at risk of progressive fibrosis (Table 4). Patients with SSc-ILD presenting with multiple predictive factors will be evaluated according to an exploratory plan including clinical examinations, PFTs, and HRCT (Figure 3a). Although the 6 MWT is not a standard monitoring tool for SSc-ILD, repeating it every 3–6 months could provide useful information. HRCT should be performed every 3 months, with the possibility of switching to annual follow-up in the absence of evidence of progression of respiratory symptoms (stable SSc-ILD). FVC and DLco measurements should be performed at least once a year. Stable patients without predictors of PPF in SSc-ILD will be monitored annually. If rapid lung function decline is detected, anti-fibrotic treatment will be started. In conclusion, clinical monitoring, PFTs, HRCT, and desaturation during exercise tests should be implemented to assess SSc-ILD progression and guide treatments.

### 4.3. Monitoring MCTD-ILD Patients for PPF Diagnosis

Monitoring of MCTD-ILD is essential, especially in patients with predictive factors for PPF phenotype (Figure 3b). Clinical examinations and measurement of FVC and DLco every 3–6 months are recommended. HRCT should be performed every 1–2 years. Although not recommended by the monitoring guidelines, the assessment of biomarkers (KL-6, SP-D) every 1–3 months is useful for assessing evolution cases who may continue to develop PPF, despite optimal treatments [197]. In those without predictive factors, follow-up is less frequent, with annual HRCT and PFTs at 12–24 months, depending on disease stability. If no symptoms or signs of deterioration occur, assessments may be performed less frequently.

### 4.4. Monitoring IIM-ILD Patients for PPF Diagnosis

Monitoring of patients with IIM-ILD should be tailored to the presence or absence of predictors for PPF. Initial evaluation includes clinical examination, HRCT to detect early signs of IIM-ILD, and PFTs (Figure 4a). The rheumatologist and pulmonologist will evaluate the patient’s symptoms and serological tests. Acute onset of respiratory symptoms, anti-MDA5 antibodies, diffuse GGO on HRCT, low ratio PaO2/FiO2 values (ratio of arterial oxygen partial pressure to fractional inspired oxygen), and multisystem symptoms (e.g., dyspnea, cough, rash, arthralgia) may indicate susceptibility to IIM-ILD progression and reduced survival [198]. The frequency of monitoring depends on the risk of fibrotic progression. In high-risk patients, PFTs should be repeated every 3–6 months to quickly detect any signs of deterioration, and HRCT is recommended annually or every 1–2 years or more often with new symptoms such as dyspnea or decreased in FVC/DLco. In patients without predictors of progression, FVC and DLco are repeated at 6–12 months and HRCT at 2–3 years.

In patients with IIM-ILD, the prognosis can be influenced by the associated conditions, and the presence of certain specific autoantibodies (MSA/MAA) is an essential indicator [199]. Testing for these antibodies is useful even in patients with IIM-ILD who do not have obvious muscle symptoms (amyopathic disease) to identify an underlying autoimmune process. Anti-MDA5 and anti-PL7 antibody positivity are correlated with higher mortality, and other antibodies (e.g., anti-TIF1γ and NXP-2) may signal an increased risk of cancer, including breast and ovarian [200].

In patients with a history of cancer, the presence of anti-TIF1γ antibody is associated with a higher risk of malignancy [201]. In cancer-associated myositis (CAM), a diagnosis of cancer is more likely within the first three years after the onset of IIM [202] and cancer screening must be tailored to individual risk, IIM subtype, and serologic profile to ensure appropriate intervention [203]. Patients at increased risk of oncological complications in IIM include those with dermatomyositis, disease onset after 40 years, persistent disease activity, severe dysphagia, skin lesions, and anti-TIF1γ and anti-NXP2 antibodies. Patients at intermediate risk of cancer include men with polymyositis, immune-mediated necrotizing myopathy (IMNM), clinically amyopathic dermatomyositis (CADM), and anti-small ubiquitin-like modifier-1 activating enzyme (anti-SAE1), anti-3-hydroxy-3-methylglutaryl coA reductase (anti-HMGCR), anti-Mi2, and anti-MDA5 antibodies. Patients stratified as low risk according to IIM subtype, autoantibody patterns, and clinical features—ASS, overlapping IIM-SARD-associated myositis, those testing positive for anti-SRP antibody, anti-Jo1 antibody, non-Jo1 ASS antibody, as well as those with myositis-associated antibodies (anti-PM-Scl, anti-Ku, anti-RNP, anti-SSA/Ro, anti-SSB/La), Raynaud’s phenomenon, arthritis, and ILD—should be better considered to be at a standard risk of IIM-related cancer. Basic cancer screening as baseline is recommended [204].

### 4.5. Monitoring SjS-ILD for PPF Diagnosis

In almost 50% of cases, SjS-ILD occurs before or concurrently with SjS diagnosis (SjS-ILD-onset), and SjS-ILD that occurs later (SjS-ILD-incident) is less studied. To date, no clinical, imaging, or functional differences have been identified between these forms [151]. Fibrosis patterns are common in SjS-ILD, but data on the onset of ILD remain incomplete [205,206]. Early detection of PPF in SjS-ILD patients is recommended to prevent respiratory failure (Figure 4b), even though the course may vary over time [207]. In asymptomatic patients or with minimal damage on PFTs or HRCT, PFT monitoring is performed every 3–6 months. Monitoring of SjS-ILD cases will be performed according to predictors for PPF (Table 4). Patients with factors predictive of PPF in SjS-ILD require repeat lung tests every 3–6 months and HRCT every 1–2 years or more frequently if signs of deterioration occur [152]. In the absence of predictors of progression, monitoring is less frequent, and if new symptoms appear, functional and imaging tests may be repeated more frequently.

### 4.6. Monitoring RA-ILD Patients for PPF Diagnosis

The potential development of PPF phenotype in 40% of RA-ILD cases necessitates periodic monitoring through clinical examination, HRCT, PFTs, 6 MWT, and exercise desaturation assessment (Figure 5a).

During the first year after diagnosis, PFTs are repeated every 3–12 months, with the frequency adjusted thereafter based on factors predictive of progression and clinical course. Patients at risk of PPF (Table 4) are monitored more frequently, undergoing clinical examinations and PFTs every 3–6 months, 6 MWT every 3–6 months, and HRCT every 6–12 months to assess fibrosis progression [18,157]. If the clinical condition worsens, HRCT can be performed at 3–6-month intervals. At 6–12 months, serological tests targeting systemic inflammation, liver and kidney functions, and fibrosis biomarkers (e.g., KL-6, SP-D, CCL18) are also recommended. For patients with RA-ILD without predictive factors of PPF, monitoring is less frequent but ongoing. Clinical examand PFTs occur at 6–12 months, and 6 MWT is performed at 6–12 months. HRCT is repeated every 2 years, but frequency may be increased if PFTs or symptoms indicate disease progression. Inflammatory markers and fibrosis-specific biomarkers are assessed every 6–12 months [208]. Stable patients can be monitored annually, but in cases of suspected progression, monitoring will be performed every 6 months.

### 4.7. Monitoring of SLE-ILD Patients for PPF Diagnosis

Monitoring of SLE-ILD is essential, especially in patients with predictive factors for a PPF phenotype (Table 4). In these patients, monitoring should be frequent and rigorous, including an initial evaluation with HRCT to detect pulmonary changes, and PFTs, including FVC and DLco (Figure 5b). Patients at high risk for PPF require repeat PFTs every 3–6 months and HRCT every 1–2 years or more often, depending on symptoms. In patients without prognostic factors, follow-up is less frequent, with HRCT repeated every 2–3 years if the initial assessment is normal, and PFTs are indicated every 12–24 months [209].

## 5. Conclusions

Screening for ILD is recommended for all patients with SARDs. Clinical and imaging evaluations with HRCT are essential diagnostic tools for early detection, while PFTs are mandatory for ongoing monitoring of SARD-ILD. The expansion of predictor panels for the progressive phenotype is challenging but also offers new perspectives for managing severe forms of SARD-ILD. Advances in novel biomarker discovery and imaging methods are expected to enhance risk stratification and improve clinical care. SARD-ILDs are complex conditions, so a comprehensive evaluation and appropriate management by a multidisciplinary team are essential to maximize the chances of improving prognosis and quality of life.

## Figures and Tables

**Figure 1 diagnostics-14-02674-f001:**
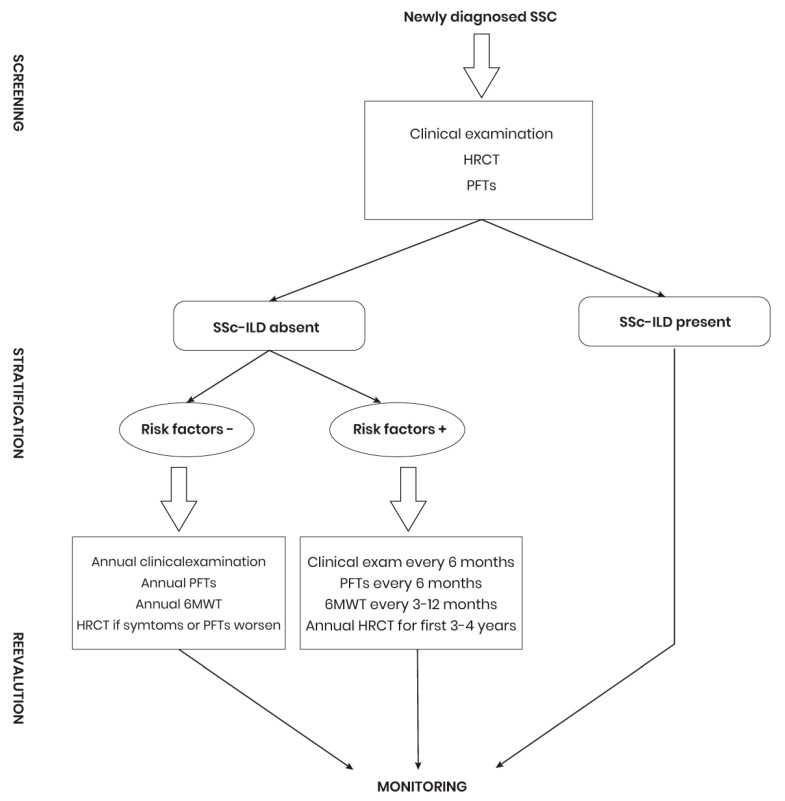
Management diagnostic plan for SSc-ILD. SSc: Systemic sclerosis; HRCT: High-resolution computed tomography; PFTs: Pulmonary function tests; SSc-ILD: Interstitial lung disease associated with systemic sclerosis; 6 MWT: Six-minute walk test.

**Figure 2 diagnostics-14-02674-f002:**
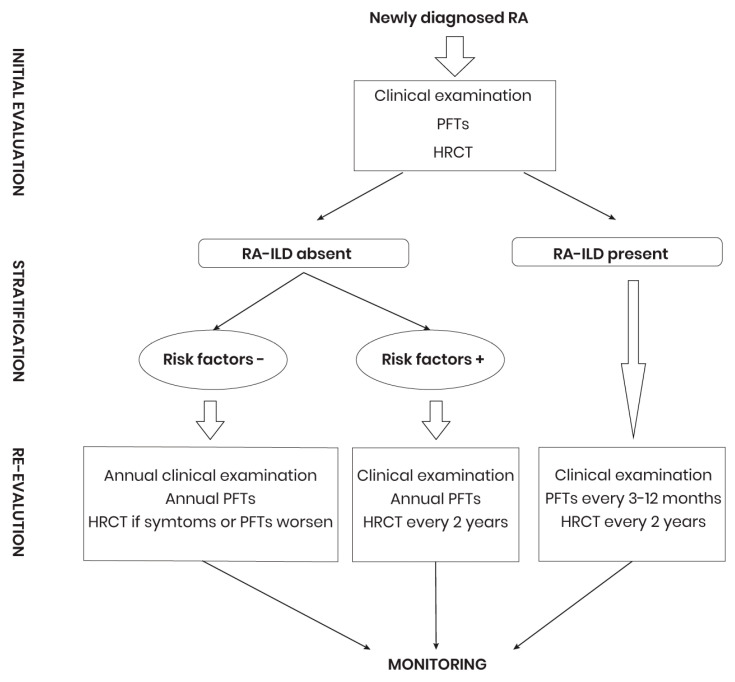
Management diagnostic plan for RA-ILD. RA: Rheumatoid arthritis; PFTs: Pulmonary function tests; RA-ILD: Rheumatoid arthritis associated interstitial lung disease; HRCT: High-resolution computed tomography.

**Figure 3 diagnostics-14-02674-f003:**
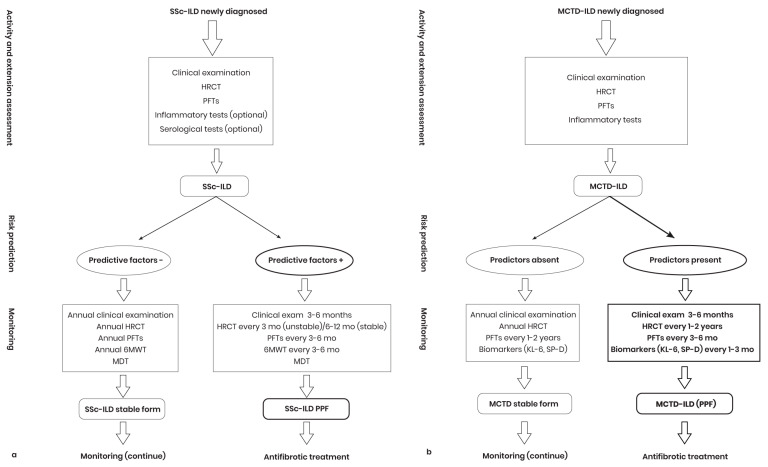
PPF phenotype early detection algorithm in SSc-ILD (**a**) and MCTD–ILD (**b**) patients. SSc-ILD: Interstitial lung disease associated with systemic sclerosis; HRCT: High-resolution computed tomography; PFTs: Pulmonary function tests; 6 MWT: Six-minute walk test; mo: Months; MDT: Multidrug therapy; MCTD-ILD: Interstitial lung disease in mixed connective tissue disease; KL-6: Krebs von den Lungen-6; SP-D: Serum surfactant protein D; PPF: Progressive pulmonary fibrosis.

**Figure 4 diagnostics-14-02674-f004:**
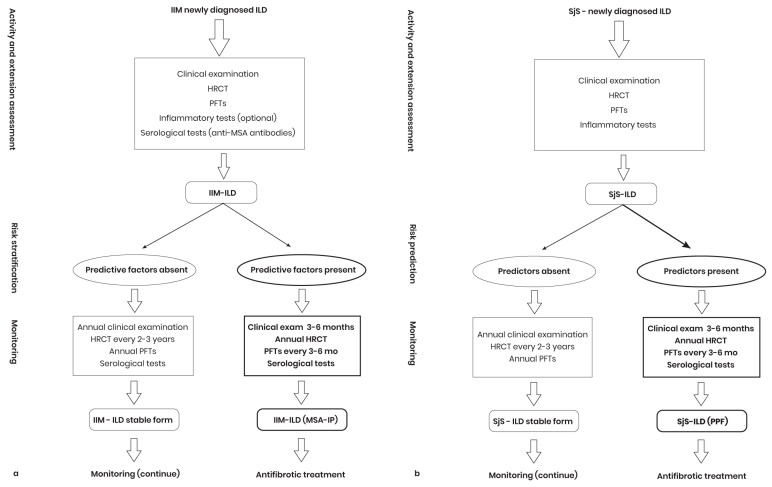
PPF early detection algorithm in IIM-ILD (**a**) and SjS-ILD (**b**) patients. IIM-ILD: Interstitial lung disease in idiopathic inflammatory myopathy; HRCT: High-resolution computed tomography; PFTs: Pulmonary function tests; anti-MSA: myositis-specific antibody; MSA-IP: myositis-specific autoantibodies–interstitial pneumonia. SjS-ILD: Interstitial lung disease associated with Sjӧgren’s syndrome; PFP: Progressive pulmonary fibrosis in interstitial lung diseases.

**Figure 5 diagnostics-14-02674-f005:**
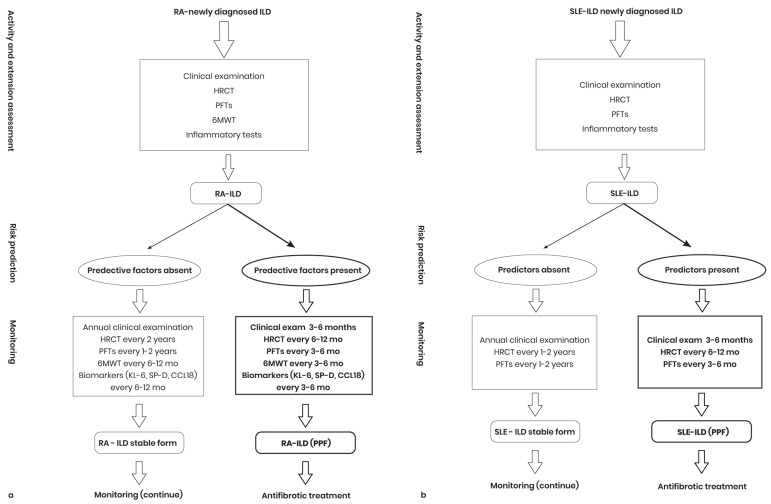
PPF early detection algorithm in RA-ILD (**a**) and SLE-ILD patients (**b**). RA-ILD: Rheumatoid-arthritis-associated interstitial lung disease; HRCT: High-resolution computed tomography; PFTs: Pulmonary function tests; 6 MWT: Six-minute walk test; mo: Months; KL-6: Krebs von den Lungen-6; SP-D: Serum surfactant protein D; CCL18: CC chemokine ligand 18; PFP-ILD: Progressive pulmonary fibrosis–interstitial lung disease; SLE-ILD: Systemic lupus-erythematosus-associated interstitial lung disease; exam: Examination; PPF: Progressive pulmonary fibrosis in interstitial lung diseases.

**Table 1 diagnostics-14-02674-t001:** The risk factors for the occurrence of SARD-ILD.

Type of SARDs	Risk Factors for SARD-ILD	References
SSc	Anti-SCL-70 and speckled antinuclear antibodies positivity	[22]
	Male gender	[23]
	African-American race	[24,25]
	Diffuse form of SSc	[24,25]
	Recently discovered disease (in the first 5–7 years)	[22]
	Increased values of acute phase reactants	[22,23]
	Anti-TRIM21 positivity	[26]
IIMs	Advanced age at diagnosis	[27]
	Arthritis/arthralgia	[27,28]
	Fever, arthritis, Raynaud’s phenomenon, skin manifestations of the “mechanic’s hands” type	[27,29]
	Elevated levels of ESR and CRP	[27,30]
	ANA positivity	[31]
	Anti-Jo-1 positivity	[30]
	Anti-MDA5 positivity	[30,32]
SjS	Male gender	[33]
	Smoking	[33]
	Older age (over 65 years)	[33]
	Long course of the disease	[33]
	Raynaud syndrome	[34]
	Increased values of ESR, CRP, lactate dehydrogenase	[35]
	Lymphopenia	[34]
	ANA and anti-Ro52/SSA positivity	[36,37]
	Anti-TRIM21 positivity	[26]
MCTD	Dysphagia	[38]
	Raynaud’s phenomenon	[39]
	Elevated C-reactive protein	[38,40]
	Anti-Ro52 and anti-Sm positivity	[38,41]
	Long life with MCTD	[41]
	Old age	[41]
	Male gender	[40,42]
	Anti-RNP elevated titer	[40,42]
RA	Male gender	[43,44,45]
	Aged over 65 years	[23,44]
	Old smokers (>10 pack-years)	[43,44,45]
	Rheumatoid nodules and erosive joint damage presence	[36,37]
	High activity of RA	[43,46,47,48]
	Functional impairment	[49]
	Long duration of the disease	[46]
	Anti-citrullinated peptide antibodies (anti-CCP) positivity	[50,51,52,53]
	Increased titers of rheumatoid factor (>100 IU/mL)	[50,52,53]
	Presence of mutations in the MUC5B gene	[54]
	Anti-TRIM21 positivity	[26]
SLE	Advanced age (over 50 years)	[41,55]
	Coexistence of another autoimmune disease (such as SjS or SSc)	[41]
	Duration of autoimmune disease (an average of 7.7 years between the onset of SLE and ILD)	[41]
	SLEDAI higher	[41]
	Association with other clinical manifestations such as Raynaud’s phenomenon or sclerodactyly in an overlap syndrome	[41,56]
	Low albumin levels	[56]
	Increased levels of acute-phase reactants	[56]
	Low level of complement	[56]
	Elevated levels of double-stranded DNA (dsDNA) antibodies	[56]
	Anti-La, anti-Scl-70 and anti-U1RNP positivity	[56]

SSc—Systemic sclerosis; Anti-SCL-70—Autoantibodies against topoisomerase; Anti-TRIM21 antibody—Tripartite motif-containing protein 21 antibody; IIMs—Idiopathic inflammatory myopathies; ESR—Erythrocyte sedimentation rate; CRP—C-reactive protein; ANA—Antinuclear antibody; Anti-JO-1 antibody—Anti-histidyl tRNA synthetase antibody; Anti-MDA5 antibody—Anti-melanoma differentiation-associated gene 5 antibody; SjS— Sjӧgren’s syndrome; Anti-Ro52/SSA antibody—Anti-Sjӧgren-syndrome-related antigen A autoantibody; MCTD—Mixed connective tissue disease; Anti-SM antibody—Antibodies against the Smith antigen; Anti-RNP antibody—Anti-ribonucleoprotein antibody; RA—Rheumatoid arthritis; Anti-CCP antibodies—anti-cyclic citrullinated peptide antibodies; RF—Rheumatoid factor; MUC5B gene—Mucin 5B, oligomeric mucus/gel-forming gene; SLE—Systemic lupus erythematosus; ILD—Interstitial lung disease; SLEDAI—Systemic lupus erythematosus disease activity index; Anti-dsDNA—anti-double stranded DNA antibodies; Anti-La antibody—anti human cytoplasmatic RNP protein complex antibody; Anti-U1RNP—Antibody to U1 ribonucleoprotein.

**Table 2 diagnostics-14-02674-t002:** Imaging (HRCT) and pathological findings in SARD-ILD.

ILD	Findings HRCT			Findings Pathological
	Characteristics	Distribution	Additional Features	Microscopic Features
NSIP [61,62,63,64]	“Ground glass” opacities, thickening of the alveolar septa.	Diffuse, predominantly in the lower lobes.	Reduction of lung volume, preservation of lung architecture, absence of hyaline membranes, and traction bronchiectasis may occur.	Lesions of alveolitis and fibrosis are uniform in age and diffusely affect the alveolar septa. Fibroblastic foci are absent, and lung architecture remains preserved. Hyaline membranes and granulomas are not typically present [65].
UIP [66]	Reticular, retractile-type opacities.	Predominantly in the lower lobes and subpleural regions.	Traction bronchiectasis and subpleural microcysts (“honeycombing”).	The pattern features alveolitis, fibrosis, fibroblastic foci, and honeycomb changes, alternating with normal lung areas. Subpleural and basal regions are the most affected. Fibrosis exceeds inflammation, with smooth muscle proliferation and type II alveolocyte hyperplasia, absent hyaline membranes [66].
COP [67]	Multiple pulmonary consolidations, usually bilateral; solitary consolidations.	Subpleural and in the lower lung fields.	“Ground glass (GGO)” lesions (frequent), annular opacities (less frequent), diffuse–infiltrative forms, minimal pleural effusion, migratory opacities.	COP is marked by granulation tissue within alveolar ducts and alveoli, with mononuclear inflammation and foamy macrophages. Fibrotic foci, called Masson corpuscles, appear as polyp-like structures. The lesions are homogeneous and maintain the overall lung architecture [67].
DIP [68,69]	“Ground glass” opacities.	Bilateral, predominantly in the lower lung fields, peripherally; may also be localized in the upper and middle lungs.	Thickening of intralobular septae, especially subpleural and basal (60–80% of cases), traction bronchiectasis, or honeycombing are rarer (25–30% of cases).	DIP typically features pigmented macrophages in alveoli, type II pneumocyte hyperplasia, and mild interstitial fibrosis. Lesions are diffuse and homogeneous, unlike UIP, which shows a mottled pattern with less uniformity [70].
LIP [71]	Reticular and nodular opacities.	Bilateral basal.	”Ground glass” lesions as the disease progresses, thin-walled cysts, 1–3 cm in diameter (60–80% of cases).	LIP features diffuse lymphoid hyperplasia affecting the lung interstitium, with lymphocytes, plasma cells, histiocytes, and reactive follicles in interalveolar septa. Advanced stages may show fibrosis and honeycombing [72].
AIP [73,74]	Areas of alveolar consolidation or “ground glass” opacities.	These changes are never subpleural or centrally localized.	The extent of the affected area correlates with the duration of the disease. Initially, ”ground glass” opacities are bilateral in focal areas and alternate with areas of unaffected lobules.	AIP, or diffuse alveolar damage, progresses rapidly through three phases: (1) the acute exudative phase (edema, hyaline membranes, inflammation), (2) the subacute proliferative phase (fibroblast proliferation, type II pneumocyte hyperplasia), and (3) chronic fibrotic phase (collagen deposits, occasional intraluminal thrombi) [75].
DAH [76]	Bilateral ground glass opacities associated with “crazy cobblestone pattern”.	Subpleural sparing is often a useful diagnostic feature.	In the subacute HRCT phase, fine diffuse nodular densities will appear, and in the later phase, there may be evidence of thickening of the interlobular septum due to intralymphatic accumulation of hemosiderin.	Diffuse intraalveolar blood admixed with hemosiderin-laden macrophages. Organizing fibroblastic tissue may also be present. In the case of capillaritis, neutrophils are seen within alveolar septa, resulting in vascular necrosis. Hyaline membranes may also be present [77].

NSIP: Non-specific interstitial pneumonia; UIP: Usual interstitial pneumonia; COP: Cryptogenic organizing pneumonia; DIP: Desquamative interstitial desquamative lung disease; LIP: Lymphocytic interstitial lung disease; AIP: Acute interstitial pneumonia; DAH: Diffuse alveolar hemorrhage.

**Table 3 diagnostics-14-02674-t003:** Main imaging (HRCT) features described in SARD-ILD.

Type of SARD-ILD	HRCT Features	Axial HRCT Images in Patients with SARD-ILD	Description of HRCT Images in Patients with SARD-ILD
SSc-ILD	NSIP, UIP with straight-edge sign and/or “four corners” sign.	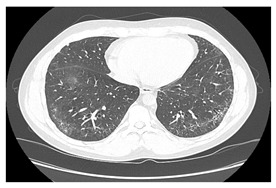	NSIP, which is the most common appearance in patients with SSc-ILD, is characterized by reticular opacities arranged in bands that run parallel to the lung’s subpleural contour, with a tendency to avoid the subpleural area.
IIM-ILD	NSIP or OP.	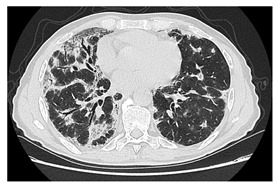	The appearance of OP is frequently encountered in IIM-ILD patients. It is characterized by ground glass areas alternating with areas of condensation, often with a perilobular distribution. OP can overlap with NSIP, leading to exacerbations of ILD, or it may present as the initial pattern.
SjS-ILD	NSIP; LIP with diffuse interstitial and peribronchiolar infiltration of lymphoplasma cells; OP.	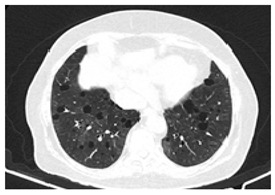	The appearance of LIP, commonly seen in patients with SjS, can be associated with ground glass lesions, nodules, septal thickenings, and cysts. The middle and lower lung fields are most frequently affected.
RA-ILD	UIP; NSIP; airway disease with obliterative and follicular bronchiolitis; rheumatoid nodules.	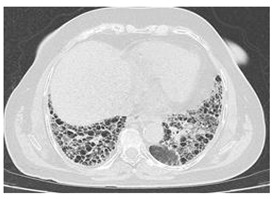	UIP, specific to RA-ILD, is characterized by traction bronchiectasis, which may present as subpleural fibrotic cysts with a “honeycomb” appearance, overlapping in multiple layers. UIP often represents the final stage in the progression of most connective tissue diseases and is associated with the most severe prognosis.
SLE-ILD	NSIP, AIP, and DAH.	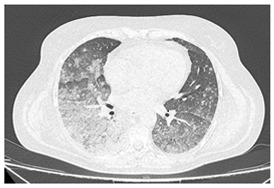	The appearance of DAH is characterized by areas of ground glass, which can evolve into consolidations due to intralobular blood accumulation. DAH is a life-threatening complication in patients with SLE-ILD.

SARD-ILD: Systemic autoimmune rheumatoid disorder associated with interstitial lung disease; HRCT: High-resolution computed tomography; SSc-ILD: Systemic sclerosis associated with interstitial lung disease; IIM-ILD: Idiopathic inflammatory myopathy associated with interstitial lung disease; SjS-ILD: Sjӧgren’s syndrome associated with interstitial lung disease; RA-ILD: Rheumatoid arthritis associated with interstitial lung disease; SLE-ILD: Systemic lupus erythematosus associated with interstitial lung disease; NSIP: Non-specific interstitial pneumonia; UIP: Usual interstitial pneumonia; OP: Organizing pneumonia; DIP: Desquamative interstitial desquamative lung disease; LIP: Lymphocytic interstitial lung disease; AIP: Acute interstitial pneumonia; DAH: Diffuse alveolar hemorrhage.

**Table 4 diagnostics-14-02674-t004:** The main predictors of PPF in ILD in patients with SARDs.

Type of SARDs	Risk Predictors for PPF	References
SSc-ILD	older age at the onset of SSc	[169,170]
	male gender	[171]
	diffuse cutaneous systemic sclerosis	[172]
	anti-Scl-70 positivity	[171,173]
	anti-RNAP3	[174]
	presence of GERD and esophageal structural changes	[175]
	high severity of ILD at the time of diagnosis more than 20% extension of fibrosis on HRCT	[176]
	reduced DLco levels	[176]
	reduced FVC levels	[177]
	decreased oxygen saturation	[178]
	elevated serum levels of some structural biomarkers (CA15-3)	[179]
	elevated serum levels of chemokines (CCL-18)	[180]
IIM-ILD	heliotropic rash	[181]
	anti-MDA5 positivity	[181]
	anti-Ro 52 positivity	[182]
SjS-ILD	male gender	[42,183]
	reticular pattern on HRCT	[42,183]
	non-sicca phenotype	[42,183]
	UIP pattern	[42,183]
MCTD-ILD	advanced age	[64]
	digital ulcers	[184]
	anti-Ro52 positivity	[185]
	NSIP pattern	[186]
RA-ILD	older age	[187]
	male gender	[187]
	smoking status	[188]
	UIP pattern or greater extension of ILD	[189]
	moderate and high values (≥3.2) of DAS28	[190,191]
	increased blood levels of KL-6	[192]
	increased titers of rheumatoid factor or anti-CCP	[193]
	low initial values of FVC and DLco	[187]
	rapid decrease of values of FVC and DLco over 6 months	[187]
SLE-ILD	current smoking status	[194]
	increased serum levels of KL-6	[194]
	projected extension of fibrosis	[194]
	NSIP plus OP patterns vs. NSIP pattern	[194]
	neuropsychiatric lupus	[194]
	thrombocytopenia	[194]
	other comorbidities of SLE	[194]
	Elevated anti-dsDNA titers	[194,195]
	Presence of overlapping syndromes	[160]

SARDs: Systemic autoimmune rheumatic diseases; SSc-ILD: Interstitial lung disease associated with systemic sclerosis; SSc: Systemic sclerosis; Anti-SCL-70: Autoantibodies against topoisomerase I; dcSSc: Diffuse cutaneous systemic sclerosis; GERD: Gastroesophageal reflux disease; ILD-SSc: Scleroderma-associated interstital lung disease; HRCT: High-resolution computed tomography; DLco: Diffusing capacity of the lungs for carbon monoxide; CA15-3: Cancer antigen 15-3; CCL-18: Chemokine (C-C motif) ligand 18; IIM-ILD: Interstitial lung disease in idiopathic inflammatory myopathy; Anti-RO52: anti-human cytoplasmatic RNP protein complex antibody; SjS-ILD: Interstitial lung disease associated with Sjӧgren’s syndrome; UIP: Usual interstitial pneumonia; MCTD-ILD: Interstitial lung disease in mixed connective tissue disease; NSIP: Non-specific interstitial pneumonia; RA-ILD: Rheumatoid arthritis associated interstitial lung disease; DAS28: Disease activity score28; KL-6: Krebs von der Lungen-6; Anti-CCP antibodies: Anti-cyclic citrullinated peptide antibodies; FVC: Forced vital capacity; SLE-ILD: Interstitial lung disease associated with systemic lupus erythematosus; OP: Organizing pneumonia; SLE: Systemic lupus erythematosus; Anti-dsDNA: Anti-double stranded DNA antibodies.

## Data Availability

Not applicable.

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
