# Peer review of "A Practical Multidisciplinary Approach to Identifying Interstitial Lung Disease in Systemic Autoimmune Rheumatic Diseases: A Clinician’s Narrative Review"

_diagnostics, 2024, doi:10.3390/diagnostics14232674_

Round 1

Reviewer 1 Report

Comments and Suggestions for Authors

 A Practical Multidisciplinary Approach to Identifying Interstitial Lung Disease in Systemic Autoimmune Rheumatic Diseases: A Clinician's Narrative Review

The manuscript covers an interesting topic ..However ,some points need to be addressed

Page

Line

Manuscript

Comments

1

30

 Each of the emblematic diseases

A confusing word .

It is better to use an easy and simple language

2

47,48

 The systemic autoimmune rheumatic diseases

The authors should follow the scientific rules of writing abbreviations

2

87

The research gap should be mentioned and should be clear and concise

3

114

The complete clinical examination will highlight the

It is not clear why the authors used italic writing for this paragraph

3

119

The 119 patient´s history will

Table 1 represents risk factors as a whole and not restricted for history only

4

122-126

The authors did not mention their references here

The authors mentioned autoantibodies in table 1 and again comment on autoantibodies in another subtitle which is to some extent confusing .

5

138

diagnostic value for interstitial lung disease

The authors again did not follow the scientific rules of writing abbreviations .

4

137

3. Imaging examination 137

Numbering of subtitles should follow the title and this is confusing

What is the difference between table 2 and table 3

B6. Bronchoalveolar lavage

What is meant by B6

9

245

C. Early diagnosis and monitoring of SARD-ILD 254

The authors used numbering with c and sometimes with 1,2 ,3 ..This is so confusing

The figures not clear and hazy

There is repetition of the same ideas and same words multiple times this makes the review redundant and not interesting to the readers .

There is some similarities between table 1 and table 4

Author Response

Dear Reviewer,

First of all, I want to thank you for your patience and great care during the review. Your advice was extremely beneficial and we tried to take it into account.

Second, I have tried to respond to your comments punctually, stating the responses in the table below. I tried to reduce the text of the article as much as possible. For this reason, subsections 3 and 4 have been completely rewritten, and four images from those originally used in Early Diagnosis have been removed.  I also merged the images in subsection 4, so out of 6 figures only 3 figures remain. I tried to enlarge the fonts in the images to make them clearer.

Thanks to you, this article has been improved, allowing as many colleagues as possible, also in specialties beyond pulmonology and rheumatology to read and study it.

Comment 1: [Each emblematic disease]

Response: We agree with this comment. Therefore, I replaced the word "emblematic" with the word "representative" (page1, line 30)

Comment 2: [Systemic autoimmune rheumatic diseases]

Response: Thank you for pointing this out. Therefore, I replaced “systemic autoimmune rheumatic diseases” with SARDs. (page 1, lines 47-48)

Comment 3: [The research gap should be mentioned and should be clear and concise]

Response: Thank you for pointing this out. Therefore, I made this the necessary clarification. I added the following phrase in parentheses „(exposure to high and repeated doses of radiation is of concern, and the utility of serum biomarkers remains commercially unfeasible)” (page 2, line 86-87)

Comment 4: [The complete clinical examination will highlight...]

Response: We agree with this comment.

Therefore, I renumbered the subsections and replaced the italics: 2.1.Clinical examination (page 3, line 115)

Comment 5: [The patient´s history will also reveal...]

Response: Thank you for pointing this out. Therefore, I would add the word “some” of the risk factors for the occurrence of SARD-ILD (Table 1). (page 3, line 123)

Comment 6: [The authors did not mention their references here]

Response: In this paragraph, in which we discuss the clinical diagnosis of patients with SARD-ILD, we did not use references.

Comment 7: [The authors mentioned autoantibodies in Table 1 and again comment on antibodies in another subtitle which is to some extent confusing]

Response: Thank you very much for pointing this out.

-In table 2, at SjS, the order in which we mention “Male gender”, references 30 and 32 will be replaced by references 33.

-In page 5, lines 132-133, references 25 and 57 will disappear.

-In page 5,  line 135, reference 33 will be replaced by reference 32.

- In page 5, line 137, references 33 and 43 will be replaced by reference 57.

Comment 8: [3.Imaging examination. The authors again did not follow the scientific rules of writing abbreviations]

Response: We agree with this comment. Therefore, we renumbered the subsections and replaced the italics: 2.3.Imaging examination (page 5, line 140)

Comment 9: [Diagnostic value for interstitial lung disease ]

Response: Thank you. We agree with this comment. Therefore, I replaced the words " interstitial lung disease " with the abbreviation “ILD” (page 5, line 141)

Comment 10: [What is the difference between table 2 and table 3?].

Response: In Table 2, we presented a summary of the imaging and pathological aspects of the main patterns found in SADR-ILD. In contrast, in Table 3 we tried to exemplify the main imaging patterns found in patients with SARD-ILD. These examples are accompanied by HRCT images that illustrate these patterns.

Comment 11: [What is means B6. Bronchoalveolar lavage]

Response: Thank you! We agree with this comment. Therefore, we renumbered the subsections as “2.6. Bronchoalveolar lavage ”

Comment 12: [C. Early diagnosis and monitoring of SARD-ILD] page 9, line 258

Response: Thank you!  We agree with this comment.

Therefore, we renumbered the section as “3. Early diagnosis and monitoring of SARD-ILD”. In this section, we have renumbered the subsections with 3.1. (page 9, line 279), 3.2. (page 10, line 313), 3.3. (page 11, line 348), 3.4. (page 13, line 385), 3.5. (page 14, line 417), 3.6. (page 15, line 442).

Next, I renamed section D through section 4 (page 16, line 468) and renumbered subsections with 4.1. (page 17, line 488), 4.2. (page 18, line 521), 4.3. (page 19, line 552), 4.4 (page 20, line 579), 4.5. (page 21, line 606), 4.6. (page 22, line 622), 4.7. (page 23, line 666).

Comment 13: [The figures are not clear and hazy]

Response: We agree with this comment. Therefore, I redid the figures using larger characters, which makes them clear. Also, I have reduced the figures in the "Early diagnosis" part, so the article can be easier to read.

Comment 14: [There is repetition of the same ideas and same words multiple times this makes the review redundant and not interesting to the readers]

Response: Thank you! We agree with these comments.

Consequently, I reduced the text of subsections 3 and 4 a lot, so that this part of the article became much easier to understand. I have significantly minimized redundant information to keep the readers engaged.

Comment 15: [There is some similarities between table 1 and table 4]

Response: Thank you very much for pointing this out.

The similarities between the two tables appear naturally because these risk/prognostic factors refer to the same pathology (disease).

Multiple studies have shown, that the same predictive factors are present for both the onset of SARD-ILD and the rapidly progressive evolution of SARD-ILD in some patients.

I think that by repeating these risk factors in both tables, I will be able to draw the readers' attention to the importance of looking for these risk/predictive factors.

I hope that in the future, our readers will recognize that taking a thorough patient history is a very important part of the clinical examination, and as a result, they will continue to allocate sufficient time to this diagnostic method.

I thank you very much for the professionalism and attention you gave to this article!

Kind regards,

Biciusca Viorel.

Reviewer 2 Report

Comments and Suggestions for Authors

In this review, Biciusca et al. provides a practical approach the diagnosis and surveillance to systemic autoimmune rheumatological disease- related interstitial lung diseases (SARD-ILD). They also review the pertinent literature. I find the review comprehensive and it provides a useful guide to these complex disease. However I have a few corrections and suggestions to improve the quality of the manuscript.

Major criticism:

1.       Manuscript length. The manuscript is too verbose, which is partially due to the significant redundance in explaining the assessment and follow up of patients by disease entity. In review of you figures 3-12 there more similarities than differences. I wonder if this could all be in one figure with pointing out the occasional differences in management.

2.       Please discuss the role of cardiac workup, including the evaluation of pulmonary hypertension ( right heart catheterization, cardiopulmonary exercise testing, echocardiogram), which is a common complicating  factor in SARD-ILD.

3.       When talking about inflammatory myositis, please discuss the autoimmune seromarkers and how they relate to the progression of ILD. There is some discussion about MDA5, but this list is more extenisve (PL antibodies, etc.). A good references would be Hallowell and Davidoff CHEST Reviews 2023;163:1476-1491 and Hannah et al. Rheumatology 2023;00:1-11.

4.       Table 2 is a nice summary of ILD imaging and pathology. However we do not usually associate DIP and RB-ILD with autoimmune disease. Please explain  or remove. Also, DAH would be a good addition to this table below AIP, as they are often related to each-other.  Please discuss the differences between NSIP and UIP in radiological appearance in Table 2 or in the text. Please list that in LIP a major finding is the few thin-walled cysts.

5.       Please discuss that cancer risk is higher than usual in SARD-ILD, especially in myositis.

6.       NSIP is not a pattern. In fact the name is non-specific interstitial pneumonia, because it does not have a pattern. Please correct.

Minor criticism:

1.       There are many typos in the test. To mention a few: Table 3, SLE-ILD, you probably mean AIP, not ALP, page 9, top row, RA-ILD, not AR-ILD. Please review.

2.       The more proper English term is blood gas analysis instead of gasometry.

3.       Page 9, Early diagnosis and monitoring, 3rd row from the bottom, what does practical difficulty mean? Is it because of difficulty to access bronchoscopy or because the risk of complication is unacceptably high? Please clarify.

Author Response

Dear Reviewer,

First of all, I want to thank you for your patience and great care during the review. Your advice was extremely beneficial and we tried to take it into account.

Second, I have tried to respond to your comments punctually, stating the responses in the table below. I tried to reduce the text of the article as much as possible. For this reason, subsections 3 and 4 have been completely rewritten, and four images from those originally used in Early Diagnosis have been removed.  I also merged the images in subsection 4, so out of 6 figures only 3 figures remain. I tried to enlarge the fonts in the images to make them clearer.

Thanks to you, this article has been improved, allowing as many colleagues as possible, also in specialties beyond pulmonology and rheumatology to read and study it.

Major criticisms

Comment 1: The length of the manuscript. The manuscript is too verbose, which is partly due to significant redundancy in explaining the assessment and follow-up of patients by disease entity. Looking at Figures 3-12, there are more similarities than differences. I wonder if this could all be in one figure, highlighting the occasional difference in management.

Response: Thank you very much for pointing this out. I tried to reduce the text of the article as much as possible. For this reason, subsections 3 and 4 have been completely rewritten, and four images from those originally used in ”Early Diagnosis” have been removed.  I also merged the images in subsection 4, so out of 6 figures only 3 figures remained.

Comment 2: Please discuss the role of cardiac workup, including evaluation of pulmonary hypertension (right heart catheterization, cardiopulmonary stress test, echocardiogram), which is a common complicating factor in SARD-ILD.

Response: Thank you for pointing this out.

I have added the suggested notions both to "Tools Diagnostic" (page 9, lines 252-257) and to ”Early diagnosis of SSc-ILD” (page 10, lines 306-310)

Comment 3: When discussing inflammatory myositis, please discuss autoimmune seromarkers and how they relate to ILD progression. There is some discussion about MDA5, but this list is more extensive (PL antibodies, etc.).  A good reference would be Hallowell and Davidoff CHEST Reviews 2023;163:1476-1491 and Hannah et al. Rheumatology 2023;00:1-11.

Response: Thank you for pointing this out. I have included the notions requested in the text of the article in the subsection "4.3. Monitoring IIM-ILD for PPF-ILD" (page 20, lines 578-590).

 The new bibliographic references can be found in the text with the numbers 202, and respective 203.

Comment 4a: Table 2 is a nice summary of the imaging and pathology of ILD. However, we do not usually associate DIP and RB-ILD with autoimmune diseases. Please explain or remove.

Response: We agree with this comment. Therefore, I made the following changes:

We kept the DIP pattern because I found a very interesting article from 2020,  that mentions the presence of this pattern in SARD-ILD (RA, SSc, SLE). ” DIP has been related to a variety of causes other than tobacco smoking, including connective tissue disease (rheumatoid arthritis, systemic sclerosis, rarely systemic erythematosus lupus) ”.Cottin V. Desquamative interstitial pneumonia: still orphan and not always benign. Eur Respir Rev. 2020, 29(156):200183. doi: 10.1183/16000617.0183-2020. PMID: 32581141; PMCID: PMC9489088

Comment 4b: However, we do not usually associate DIP and RB-ILD with autoimmune diseases. Please explain or remove.

Response: We excluded RB-ILD, to avoid any further comment, because this aspect is present almost exclusively in smokers.

Comment 4c: DAHs would also be a good addition to this table below AIPs, as they are often related to each other.

Response: We added the DAH-type pattern, which is found in SARD-ILD. I made the changes in Table 2.

Comment 4d: Please discuss the differences between NSIP and UIP in radiological appearance in Table 2 or in the text.

Response : We agree with this comment. Therefore, we added the comment in which I mentioned the imaging differences between NSIP and UIP: page 6, lines 157-163

Comment 4e: Please list that in LIP a major finding are some thin walled cysts.

Response: Thank you for pointing this out!Therefore, we modified the text, where I mentioned these characteristics.

Comment 5: Please discuss that the risk of cancer is higher than usual in SARD-ILD, especially in myositis.

Response: I have included the requested notation in the text of the article in the subsection "4.3. Monitoring IIM-ILD for PPF-ILD" (page 20, lines 590-603). The new bibliographic references can be found in the text with the numbers: 204-207 (in 4.4. Monitoring IIM-ILD patients for PPF diagnosis, page 16-17)

Comment 6: NSIP is not a model. In fact, the name is nonspecific interstitial pneumonia because it does not have a pattern. Please correct.

Response: In response to this comment, we would like to request more explicit guidance.

NSIP is an imaging and pathological pattern observed in various diseases. While it can indicate stable sinus disease, in the context of systemic autoimmune rheumatic diseases (SARDs), it is considered an imaging syndrome. How can we define NSIP more clearly to preempt further comments?

How could we define it, to avoid further comments?

Minor criticisms

Comment 7: There are many typos in the test. To mention a few: Table 3, SLE-ILD, you probably mean AIP, not ALP, page 9, top row, RA-ILD, not AR-ILD. Please review.

Response: Thank you very much!  We have corrected all the mistakes.

Comment 8: The more appropriate English term is blood gas analysis instead of gasometry.

Response: Thank you! I replaced the term gasometry with arterial blood gas analysis (ABGA) in page 7, line 188.

Comment 9: Page 9, Early diagnosis and monitoring, 3rd row from the bottom, what does practical difficulty mean? Because of the difficulty of access to bronchoscopy or because the risk of complication is unacceptably high? Please clarify.

Response: Therefore we excluded the paragraph related to the practical difficulty of bronchoscopy (page 9, line 275).

Kind regards,

Biciusca Viorel.

Round 2

Reviewer 1 Report

Comments and Suggestions for Authors

improved 

Author Response

Dear Reviewer,

Thank you very much for your kindness in accepting our article review. We appreciate the exceptional quality of your expertise and welcome suggestions. We are considering all your indications, as I specified in the first answer.

In the actual version, we have corrected and underlined with the blue color code.

We appreciate that the current form of the article is much improved and provides a clear picture of the subject. We consider that the proposed objectives have been achieved.

We wish you continued success!

With great respect,

Viorel Bicusca et al.

Reviewer 2 Report

Comments and Suggestions for Authors

Dear Authors,

Thank you for the nice revision. The manuscript is now easier to follow and contains all major updates. Regarding NSIP, I continue to think calling it a pattern will cause confusion and unnecessary criticism. I see it in other papers as well, but it does not correspond to its original description. I recommend that you describe it as radiological finding. For reference, please use: Travis et al. Idiopathic Nonspecific Interstitial Pneumonia  AJRRCM 2008;177:1338-1347.

Author Response

Dear Reviewer

Thank you very much for your kindness in accepting our article review. We appreciate the special quality of the expertise and the welcome suggestions.

First of all, exploring the literature I found that there are many opinions on this topic and we will consider your suggestion. In this sense, I eliminated the term "pattern" replacing it with findings both in the text and in the two tables, as you very well suggested.

In the actual version, we have corrected and underlined with the blue color code.

Travis et al's study, which states that idiopathic NSIP represents a distinct clinical entity characterized by specific clinical, radiological, and pathological features, different from other idiopathic interstitial pneumopathies (IPPs), refers only to idiopathic NSIP. Extrapolating these notions in the case of SARD-ILD, we tried to capture the clinical, imaging, and pathological features of SARD-associated ILD.

Next, we wish you much success in your work,

With great respect,

Biciusca Viorel et al.
